# Documents as multiple overlapping windows into a grid of counts

**Alessandro Perina**[1]        **Nebojsa Jojic**[1]        **Manuele Bicego**[2]        **Andrzej Turski**[1]

[1]*Microsoft Corporation, Redmond, WA*
[2]*University of Verona, Italy*

## Abstract

In text analysis documents are often represented as disorganized bags of words; models of such count features are typically based on mixing a small number of topics [1, 2]. Recently, it has been observed that for many text corpora documents evolve into one another in a smooth way, with some features dropping and new ones being introduced. The counting grid [3] models this spatial metaphor literally: it is a grid of word distributions learned in such a way that a document's own distribution of features can be modeled as the sum of the histograms found in a window into the grid. The major drawback of this method is that it is essentially a mixture and all the content must be generated by a single contiguous area on the grid. This may be problematic especially for lower dimensional grids. In this paper, we overcome this issue by introducing the *Componential Counting Grid* which brings the componential nature of topic models to the basic counting grid. We evaluated our approach on document classification and multimodal retrieval obtaining state of the art results on standard benchmarks.

## 1   Introduction

A collection of documents, each consisting of a disorganized bag of words is often modeled compactly using mixture or admixture models, such as Latent Semantic Analysis (LSA) [4] and Latent Dirichlet Allocation (LDA) [1]. The data is represented by a small number of semantically tight topics, and a document is assumed to have a mix of words from an even smaller subset of these topics. There are no strong constraints in how the topics are mixed [5].

Recently, an orthogonal approach emerged: it has been observed that for many text corpora documents evolve into one another in a smooth way, with some words dropping and new ones being introduced. The counting grid model (CG) [3] takes this spatial metaphor – of moving through sources of words and dropping and picking new words – literally: it is multidimensional grid of word distributions, learned in such a way that a document's own distribution of words can be modeled as the sum of the distributions found in some window into the grid. By using large windows to collate many grid distributions from a large grid, CG model can be a very large mixture without overtraining, as these distributions are highly correlated. LDA model does not have this benefit, and thus has to deal with a smaller number of topics to avoid overtraining.

In Fig.1a we show an excerpt of a grid learned from cooking recipes from around the world. Each position in the grid is characterized by a distribution over the words in a vocabulary and for each position we show the 3 words with higher probability whenever they exceed a threshold. The shaded positions, are characterized by the presence, with a non-zero probability, of the word "bake"[1]. On the grid we also show the windows $\mathbf{W}$ of size $4 \times 5$ for 5 recipes. *Nomi* (1), an Afghan egg-based bread, is close to the recipe of the usual *pugliese bread* (2), as indeed they share most of the ingredients and procedure and their windows largely overlap. Note how moving from (1) to (2) the word

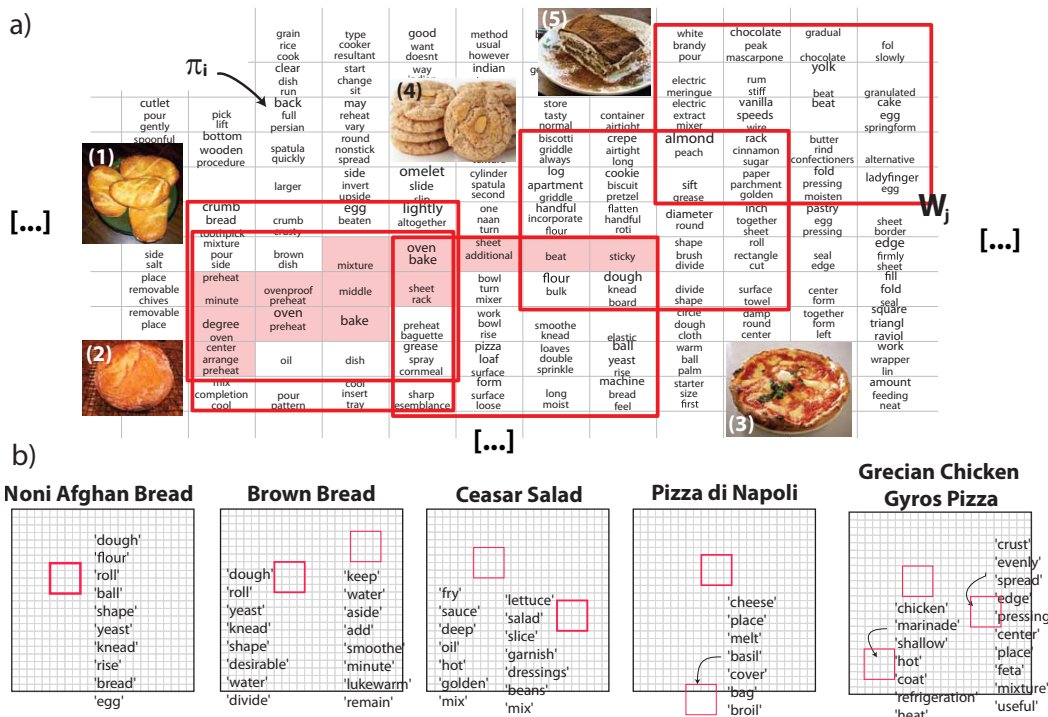

Figure 1: a) A *particular* of a $\mathbf{E} = 30 \times 30$ componential counting grid $\pi_\mathbf{i}$ learned over a corpus of recipes. In each cell we show the 0-3 most probable words greater than a threshold. The area in shaded red has $\pi('bake') > 0$. b) For 6 recipes, we show how their components are mapped onto this grid. The "mass" of each component (e.g., $\theta$ see Sec.2) is represented with the window thickness. For each component $c = \mathbf{j}$ in position $\mathbf{j}$, we show the words generated in each window $c_z \cdot \sum_{\mathbf{j} \in W_i} \pi_\mathbf{j}(z)$

"egg" is dropped. Moving to the right we encounter the *basic pizza* (3) whose dough is very similar to the bread's. Continuing to the right words often associated to desserts like sugar, almond, etc emerge. It is not surprising that baked desserts such as *cookies* (4), and pastry in general, are mapped here. Finally further up we encounter other desserts which do not require baking, like *tiramisu* (5), or *chocolate crepes*. This is an example of a "topical shift"; others appear in different portions of the full grid which is included in the additional material.

The major drawback of counting grids is that they are essentially a mixture model, assuming only *one* source for all features in the bag and the topology of the space highly constrains the document mappings resulting in local minima or suboptimal grids. For example, more structured recipes like *Grecian Chicken Gyros Pizza* or *Tex-Mex pizza* would have very low likelihood, as words related to meat, which is abundant in both, are hard to generate in the baking area where the recipes would naturally goes.

As first contribution we extend here the counting grid model so that each document can be represented by multiple latent windows, rather than just one. In this way, we create a substantially more flexible admixture model, the componential counting grid (CCG), which becomes a direct generalization of LDA as it does allow multiple sources (e.g., the windows) for each bag, in a mathematically identical way as LDA. But, the equivalent of LDA topics are windows in a counting grid, which allows the model to have a very large number of topics that are highly related, as shift in the grid only slightly refines any topic.

Starting from the same grid just described, we recomputed the mapping of each recipe which now can be described by multiple windows, if needed. Fig. 1b shows mappings for some recipes. Also the words generated in each component are shown. The three pizzas place most of the mass in the same area (dough), but the words related to the topping are borrowed from different areas. Another example is the *Caesar salad* which have a component in the salad/vegetable area, and borrows the

croutons from the bread area.

By observing Fig.1b, one can also notice how the embedding produced by CCGs yields to a similarity measure based on the grid usage of each sample. For example, words relative to the three pizzas are generated from windows that overlap, therefore they share words usage and thus they are "similar". As second contribution we exploited this fact to define a novel generative kernel, whose performance largely outperformed similar classification strategies based on LDA's topic usage [1,2]. We evaluated componential counting grids and in particular the kernel, on the 20-Newsgroup dataset [6], on a novel dataset of recipes which we will make available to the community, and on the recent "Wikipedia picture of the day" dataset [7]. In all the experiments, CCGs set a new state of the art. Finally, for the first time we explore visualization through examples and videos available in the additional material.

## 2 Counting Grids and Componential Counting Grids

The basic Counting Grid $\pi_{\mathbf{i}}$ is a set of distributions over the vocabulary on the $N$-dimensional discrete grid indexed by $\mathbf{i}$ where each $i_d \in [1 \ldots E_d]$ and $\mathbf{E}$ describes the extent of the counting grid in $d$ dimensions. The index $z$ indexes a particular word in the vocabulary $z = [1 \ldots Z]$ being $Z$ the size of the vocabulary. For example, $\pi_{\mathbf{i}}('Pizza')$ is the probability of the word "Pizza" at the location $\mathbf{i}$. Since $\pi$ is a grid of distributions, $\sum_z \pi_{\mathbf{i}}(z) = 1$ everywhere on the grid. Each bag of words is represented by a list of words $\{\mathbf{w}^t\}_{t=1}^T$ and each word $w_n^t$ takes a value between $1$ and $Z$. In the rest of the paper, we will assume that all the samples have $N$ words.

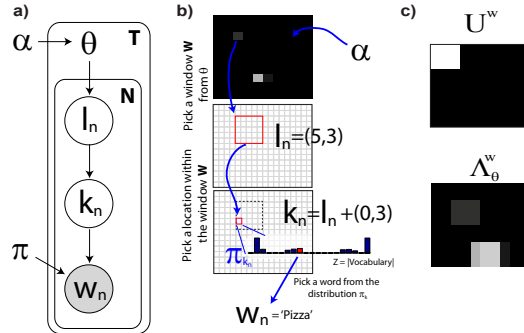

Figure 2: a) Plate notation representing the CCG model. b) CCG generative process for one word: Pick a window from $\theta$, Pick a position within the window, Pick a word. c) Illustration of $U^W$ and $\Lambda_\theta^W$ relative to the particular $\theta$ shown in plate *b)*.

Counting Grids assume that each bags follow a word distribution found somewhere in the counting grid; in particular, using windows of dimensions $\mathbf{W}$, a bag can be generated by first averaging all counts in the window $W_{\mathbf{i}}$ starting at grid location $\mathbf{i}$ and extending in each direction $d$ by $W_d$ grid positions to form the histogram $h_{\mathbf{i}}(z) = \frac{1}{\prod_d W_d} \sum_{\mathbf{j} \in W_{\mathbf{i}}} \pi_{\mathbf{j}}(z)$, and then generating a set of features in the bag (see Fig.1a where we used a $3 \times 4$ window). In other words, the position of the window $\mathbf{i}$ in the grid is a latent variable given which we can write the probability of the bag as

$$p(\{\mathbf{w}\}|\mathbf{i}) = \prod_n h_{\mathbf{i},z} = \prod_n \left(\frac{1}{\prod_d W_d} \cdot \sum_{\mathbf{j} \in W_{\mathbf{i}}} \pi_{\mathbf{j}}(w_n)\right),$$

Relaxing the terminology, $\mathbf{E}$ and $\mathbf{W}$ are referred to as, respectively, the counting grid and the window size. The ratio of the two volumes, $\kappa$, is called the capacity of the model in terms of an *equivalent number of topics*, as this is how many non-overlapping windows can be fit onto the grid. Finally, with $W_{\mathbf{i}}$ we indicate the particular window placed at location $\mathbf{i}$.

**Componential Counting Grids** As seen in the previous section, counting grids generate words from a distribution in a window $W$, placed at location $\mathbf{i}$ in the grid. Windows close in the grid generate similar features because they share many cells: As we move the window on the grid, some new features appear while others are dropped. On the other hand componential models, like [1], represent the standard way of modeling of text corpora. In these models each feature can be generated by a different source or topic, and documents are then seen as admixtures of topics. Componential counting grids get the best of both worlds: being based on the counting grid geometry they capture smooth shifts of topics, plus their componential nature, which allows documents to be generated by several windows (akin to LDA's topics). The number of windows need not be specified a-priori.

Componential Counting Grids assumes the following generative process (also illustrated by Fig.2b.) for each document in a corpus:

1. Sample the multinomial over the locations $\theta \sim Dir(\alpha)$
2. For each of the N words $w_n$
   a) Choose a at location $l_n \sim Multinomial(\theta)$ for a window of size $\mathbf{W}$
   b) Choose a location within the window $\mathbf{W}_{l_n}$; $k_n$
   c) Choose a word $w_n$ from $\pi_{k_n}$

As visible, each word $w_n$ is generated from a different window, placed at location $l_n$, but the choice of the window follows the same prior distributions $\theta$ for all words. It worth noticing that when $\mathbf{W} = 1 \times 1$, $l_n = k_n$ and the model becomes Latent Dirichlet Allocation.

The Bayesian network is shown in Fig.2a) and it defines the following joint probability distribution

$$P = \prod_{t,n} \sum_{l_n} \sum_{k_n} p(w_n|k_n, \pi) \cdot p(k_n|l_n) \cdot p(l_n|\theta) \cdot p(\theta|\alpha) \tag{1}$$

where $p(w_n = z|k_n = \mathbf{i}, \pi) = \pi_{\mathbf{i}}(z)$ is a multinomial over the vocabulary, $p(k_n = \mathbf{i}|l_n = \mathbf{k}) = U^W(\mathbf{i} - \mathbf{k})$ is a distribution over the grid locations, with $U^W$ uniform and equal to $(\frac{1}{|W|})$ in the upper left window of size $\mathbf{W}$ and 0 elsewhere (See Fig.2c). Finally $p(l_n|\theta) = \theta(l)$ is the prior distribution over the windows location, and $p(\theta|\alpha) = Dir(\theta; \alpha)$ is a Dirichlet distribution of parameters $\alpha$.

Since the posterior distribution $p(\mathbf{k}, \mathbf{l}, \theta|\mathbf{w}, \pi, \alpha)$ is intractable for exact inference, we learned the model using variational inference [8].

We firstly introduced the posterior distributions $q$, approximating the true posterior as $q^t(\mathbf{k}, \mathbf{l}, \theta) = q^t(\theta) \cdot \prod_n \left( q^t(k_n) \cdot q^t(l_n) \right)$ being $q(k_n)$ and $q(l_n)$ multinomials over the locations, and $q(\theta)$ a Dirac function centered at the optimal value $\hat{\theta}$.

Then by bounding (variationally) the non-constant part of $\log P$, we can write the negative free energy $\mathcal{F}$, and use the iterative variational EM algorithm to optimize it.

$$\log P \geq -\mathcal{F} = \sum_t \left( \sum_n \left( \sum_{l_n, k_n} q^t(k_n) \cdot q^t(l_n) \cdot \log \pi_{k_n}(w_n) \cdot U^W(k_n - l_n) \cdot \theta_{l_n} \cdot p(\theta|\alpha) \right) - \mathbb{H}(q^t) \right)$$

$$\tag{2}$$

where $\mathbb{H}(q)$ is the entropy of the distribution $q$.

Minimization of Eq. 2 reduces in the following update rules:

$$q^t(k_n = \mathbf{i}) \quad \propto \quad \pi_{\mathbf{i}}(w_n) \cdot \exp\left( \sum_{l_n=\mathbf{j}} q^t(l_n = \mathbf{j}) \cdot \log U^W(\mathbf{i} - \mathbf{j}) \right) \tag{3}$$

$$q^t(l_n = \mathbf{i}) \quad \propto \quad \theta^t(\mathbf{i}) \cdot \exp\left( \sum_{k_n=\mathbf{j}} q^t(k_n = \mathbf{j}) \cdot \log U^W(\mathbf{j} - \mathbf{i}) \right) \tag{4}$$

$$\theta^t(\mathbf{i}) \quad \propto \quad \alpha_{\mathbf{i}} - 1 + \sum_n q^t(l_n = \mathbf{i}) \tag{5}$$

$$\pi_{\mathbf{i}}(z) \quad \propto \quad \sum_t \sum_n q^t(k_n = \mathbf{i})^{[w_n=z]} \tag{6}$$

where $[w_n = z]$ is an indicator function, equal to 1 when $w_n$ is equal to $z$. Finally, the parameters $\alpha$ of the Dirichlet prior can be either kept fixed [9] or learned using standard techniques [10].

The minimization procedure described by Eqs.3-6 can be carried out efficiently in $\mathcal{O}(N \log N)$ time using FFTs [11].

Some simple mathematical manipulations of Eq.1 can yield to a speed up. In fact, from Eq.1 one can marginalize the variable $l_n$

$$
\begin{aligned}
P &= \prod_{t,n} \sum_{l_n=\mathbf{i}, k_n=\mathbf{j}} p(w_n|k_n = \mathbf{j}) \cdot p(k_n = \mathbf{j}|l_n = \mathbf{i}) \cdot p(l_n = \mathbf{i}|\theta) \cdot p(\theta|\alpha) \\
&= \prod_{t,n} \sum_{l_n=\mathbf{i}, k_n=\mathbf{j}} \pi_{\mathbf{j}}(w_n) \cdot U^W(\mathbf{j} - \mathbf{i}) \cdot \theta(\mathbf{i}) \cdot p(\theta(\mathbf{i})|\alpha_{\mathbf{i}}) \\
&= \prod_{t,n} \sum_{k_n=\mathbf{j}} \pi_{\mathbf{j}}(w_n) \cdot \left( \sum_{l_n=\mathbf{i}} U^W(\mathbf{j} - \mathbf{i}) \cdot \theta(\mathbf{i}) \right) \cdot p(\theta(\mathbf{i})|\alpha_{\mathbf{i}}) = \prod_{t,n} \sum_{k_n=\mathbf{j}} \pi_{\mathbf{j}}(w_n) \cdot \Lambda^W_{\theta^t} \cdot p(\theta(\mathbf{i})|\alpha)
\end{aligned}
\tag{7}
$$

where $\Lambda_\theta^W$ is a distribution over the grid locations, equal to the convolution of $U^W$ with $\theta$. The update for $q(k)$ becomes

$$q^t(k_n = \mathbf{i}) \propto \pi_{\mathbf{i}}(w_n) \cdot \Lambda_\theta^W(\mathbf{i}) \tag{8}$$

In the same way, we can marginalize the variable $k_n$

$$P = \prod_{t,n} \sum_{l_n=\mathbf{i}} \theta(\mathbf{i}) \cdot \Big( \sum_{k_n=\mathbf{j}} U^W(\mathbf{j}-\mathbf{i}) \cdot \pi_{\mathbf{j}}(w_n) \Big) \cdot p(\theta(\mathbf{i})|\alpha_{\mathbf{i}}) = \prod_{t,n} \sum_{l_n=\mathbf{i}} \theta(\mathbf{i}) \cdot h_{\mathbf{i}}(w_n) \cdot p(\theta(\mathbf{i})|\alpha_{\mathbf{i}}) \tag{9}$$

to obtain the new update for $q^t(l_n)$

$$q^t(l_n = \mathbf{i}) \propto h_{\mathbf{i}}(w_n) \cdot \theta^t(\mathbf{i}) \tag{10}$$

where $h_{\mathbf{i}}$ is the feature distribution in a window centered at location $\mathbf{i}$, which can be efficiently computed in linear time using cumulative sums [3]. Eq.10 highlights further relationships between CCGs and LDA: CCGs can be thought as an LDA model whose topics live on the space defined by the counting grids geometry. The new updates for the cell distribution $q(k)$ and the window distribution $q(l)$, require only a single convolution and, more importantly, they don't directly depend on each other. The model becomes more efficient and has a faster convergence. This is very critical especially when we are analyzing big text corpora.

The most similar generative model to CCG comes from the statistic community. Dunson et al. [12] worked on sources positioned in a plane at real-valued locations, with the idea that sources within a radius would be combined to produce topics in an LDA-like model. They used an expensive sampling algorithm that aimed at moving the sources in the plane and determining the circular window size. The grid placement of sources of CCG yields much more efficient algorithms and denser packing.

## 2.1 A Kernel based on CCG embedding

Hybrid generative discriminative classification paradigms have been shown to be a practical and effective way to get the best of both worlds in approaching classification [13–15]. In the context of topic models a simple but effective kernel is defined as the product of the topic proportions of each document. This kernel measures the similarity between topic usage of each sample and it proved to be effective on several tasks [15–17]. Despite CCG's $\theta$s, the locations proportions, can be thought as LDA's, we propose another kernel, which exploits exactly the same geometric reasoning of the underlying generative model. We observe in fact that by construction, each point in the grid depends by its neighborhood, defined by $\mathbf{W}$ and this information is not captured using $\theta$, but using $\Lambda_\theta^W$ which is defined by spreading $\theta$ in the appropriate window (Eq.7).

More formally, given two samples $t$ and $u$, we define a kernel based on CCG embedding as

$$K(t,u) = \sum_{\mathbf{i}} \mathcal{S}(\Lambda_{\theta^t}^W(\mathbf{i}), \Lambda_{\theta^u}^W(\mathbf{i})) \quad where \quad \Lambda_\theta^W(\mathbf{i}) = \sum_{\mathbf{j}} U^W(\mathbf{i}-\mathbf{j}) \cdot \theta(\mathbf{j}) \tag{11}$$

where $\mathcal{S}(\cdot,\cdot)$ is any similarity measure which defines a kernel.
In our experiments we considered the simple product, even if other measures, such as histogram intersection can be used. The final kernel turns to be ($\times$ is the dot-product)

$$K_{LN}(t,u) = \sum_{\mathbf{i}} \Lambda_{\theta^t}^W(\mathbf{i}) \cdot \Lambda_{\theta^u}^W(\mathbf{i}) = Tr\big(\Lambda_{\theta^t}^W \times \Lambda_{\theta^u}^W\big) \tag{12}$$

## 3 Experiments

Although our model is fairly simple, it is still has multiple aspects that can be evaluated. As a generative model, it can be evaluated in left-out likelihood tests. Its latent structure, as in other generative models, can be evaluated as input to classification algorithms. Finally, as both its parameters and the latent variables live in a compact space of dimensionality and size chosen by the user, our learning algorithm can be evaluated as an embedding method that yields itself to data visualization applications. As the latter two have been by far the more important sets of metrics when it comes to real-world applications, our experiments focus on them.

In all the tests we considered squared grids of size $\mathbf{E} = [\mathbf{40 \times 40}, \mathbf{50 \times 50}, \dots, \mathbf{90 \times 90}]$ and windows of size $\mathbf{W} = [\mathbf{2 \times 2}, \mathbf{4 \times 4}, \dots, \mathbf{8 \times 8}]$. A variety of other methods are occasionally compared to, with slightly different evaluation methods described in individual subsections, when appropriate.

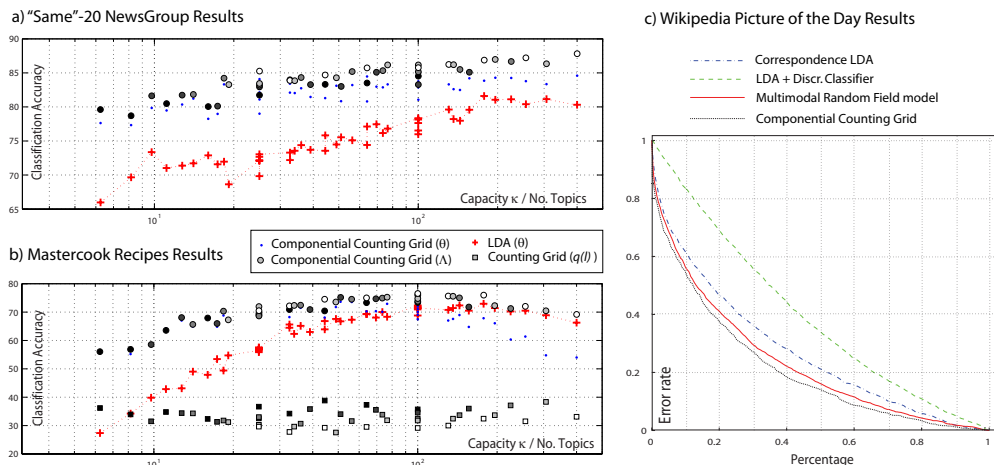

Figure 3: a-b) Results for the text classification tasks. The Mastercook recipes dataset is available on www.alessandroperina.com. We represented the grid size **E** using gray levels (see the text). c) Wikipedia Picture of the day result: average Error rate as a function of the percentage of the ranked list considered for retrieval. Curves closer to the axes represents better performances.

**Document Classification**    We compared componential counting grids (CCGs) with counting grids [3] (CGs), latent Dirichlet allocation [1] (LDA) and the spherical admixture model [2] (SAM), following the validation paradigm previously used in [2, 3].

Each data sample consists of a bag of words and a label. The bags were used without labels to train a model that capture covariation in word occurrences, with CGs mostly modeling thematic shifts, LDA and SAM modeling topic mixing and CCGs both aspects. Then, the label prediction task is performed in a 10-folds crossevaluation setting, using the linear kernel presented in Eq.12 which for LDA reduces in using a linear kernel on the topic proportions. To show the effectiveness of the spreading in the kernel definition, we also report results by employing CCG's $\theta$s instead of $\Lambda_\theta^W$. For CGs we used the original strategy [3], Nearest Neighbor in the embedding space, while for SAM we reported the results from the original paper. To the best of our knowledge the strategies just described, based on [3] and [2], are two of the most effective methods to classify text documents. SAM is characterized by the same hierarchical nature of LDA, but it represents bags using directional distributions on a spherical manifold modeling features frequency, presence and absence. The model captures fine-grained semantic structure and performs better when small semantic distinctions are important. CCGs map documents on a probabilistic simplex (e.g., $\theta$) and for $\mathbf{W} > [1 \times 1]$ can be thought as an LDA model whose topics, $h_{\mathbf{i}}$, are much finer as computed from overlapping windows (see also Eq.10); a comparison is therefore natural.

As first dataset we considered the CMU newsgroup dataset[2]. Following previous work [2, 3, 6] we reduced the dataset into subsets with varying similarities among the news groups; **news-20-different**, with posts from rec.sport.baseball, sci.space and alt.atheism, **news-20-similar**, with posts from rec.talk.baseball, talk.politics.gun and talk.politics.misc and **news-20-same**, with posts from comp.os.ms-windows, comp.windows.x and comp.graphics. For the **news-20-same** subset (the hardest), in Fig.3a we show the accuracies of CCGs and LDA across the complexities. On the x-axis we have the different model size, in term of capacity $\kappa$, whereas in the y-axis we reported the accuracy. The same $\kappa$ can be obtained with different choices of **E** and **W** therefore we represented the grid size **E** using gray levels, the lighter the marker the bigger the grid. The capacity $\kappa$ is roughly equivalent to the number of LDA topics as it represents the number of independent windows that can be fit in the grid and we compared the with LDA using this parallelism [18].

Componential counting grids outperform Latent Dirichlet Allocation across all the spectrum and the accuracy regularly raises with $\kappa$ independently from the Grid size[3]. The priors helped to prevent overtraining for big capacities $\kappa$. When using CCG's $\theta$s to define the kernel, as expected the accu-

Table 1: **Document classification**. The improvement on `Similar` and `Same` are statistically significant. The accuracies for SAM are taken from [2] and they represent the best results obtained across the choice of number of topics. BOW stands for classification with a linear SVM on the counts matrices.

| Dataset | CCG | 2D CG [3] | 3D CG [3] | LDA [1] | BOW | SAM* [2] |
|---|---|---|---|---|---|---|
| Different | **96,49**% | **96,51**% | **96,34**% | 91,8% | 91,43% | 94,1% |
| Similar | **92,81**% | 89,72% | 90,11% | 85,7% | 81,52% | 88,1% |
| Same | **83,63**% | 81,28% | 81,03% | 75,6% | 71,81% | 78,1% |

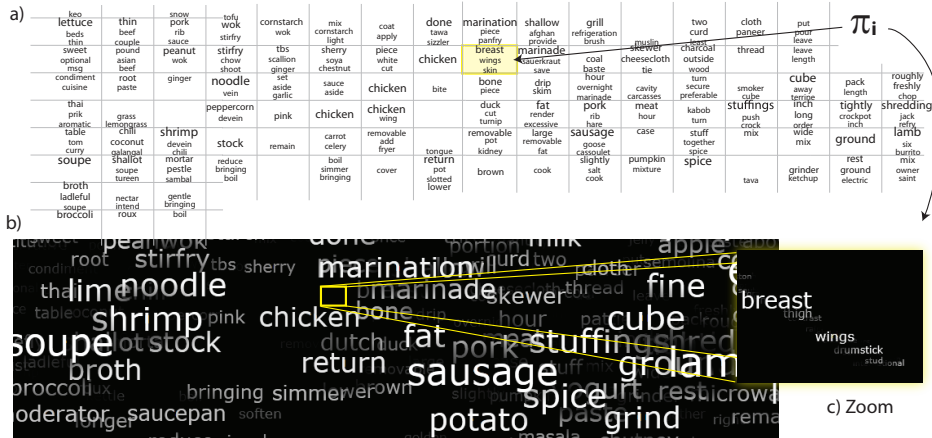

Figure 4: A simple interface built upon the word embedding $\pi$.

racy dropped (blue dots in Fig.3).

Results for all the datasets and for a variety of methods, are reported in Tab.1 where we employed 10% of the training data as validation set to pick a complexity (a different complexity have been chosen for each fold). As visible, CCG outperforms other models, with a larger margin on the more challenging **same** and **similar** datasets, where we would indeed expect that quilting the topics to capture fine-grained similarities and differences would be most helpful.

As second dataset, we downloaded 10K Mastercook recipes, which are freely available on the web in plain text format. Then we extracted the words of each recipe from its ingredients and cooking instructions and we used the origin of the recipe, to divide the dataset in 15 classes[4]. The resulting dataset has a vocabulary size of 12538 unique words and a total of $\sim$1M tokens.

To classify the recipes we used 10-fold crossevaluation with 5 repetitions, picking 80 random recipes per-class for each repetition. Classification results are illustrated in Fig. 3b. As for the previous test, CCG classification accuracies grows regularly with $\kappa$ independently from the grid size $\mathbf{E}$. Componential models (e.g., LDA and CCGs) performed significantly better as to correctly classify the origin of a recipe, spice palettes, cooking style and procedures must be identified. For example while most Asian cuisines uses similar ingredients and cooking procedures they definitely have different spice palettes. Counting Grids, being mixtures, cannot capture that as they map a recipe in a single location which heavily depends on the ingredients used. Among componential models, CCGs work the best.

**Multimodal Retrieval** We considered the *Wikipedia Picture of the Day dataset* [7], where the task is multi-modal image retrieval: given a text query, we aim to find images that are most relevant to it. To accomplish this, we firstly learned a model using the *visual* words of the training data $\{\mathbf{w}^{t,V}\}$, obtaining $\theta^t, \pi_{\mathbf{i}}^V$. Then, keeping $\theta^t$ fixed and iterating the M-step, we embedded the textual words $\{\mathbf{w}^{t,T}\}$ obtaining $\pi_{\mathbf{i}}^W$. For each test sample we inferred the values of $\theta^{t,V}$ and $\theta^{t,W}$ respectively from $\pi_{\mathbf{i}}^V$ and $\pi_{\mathbf{i}}^W$ and we used Eq.12 to compute the retrieval scores. As in [7] we split the data in 10

folds and we used a validation set to pick a complexity. Results are illustrated in Fig.3c. Although we used this simple procedure without directly training a multimodal model, CCGs outperform LDA, CorrLDA [19] and the multimodal document random field model presented in [7] and sets a new state of the art.

The area under the curve (AUC) for our method is $21.92 \pm 0.6$, while for [7] is $23.14 \pm 1.49$ (Smaller values indicate better performance). Counting Grids and LDA both fail with AUCs around 40.

**Visualization** Important benefits of CCGs are that 1) they lay down sources $\pi_{\mathbf{i}}$ on a 2-D dimensional grid, which are ready for visualization, and 2) they enforce that close locations generate similar topics, which leads to smooth thematic shifts that provide connections among distant topics on the grid. This is very useful for *sensemaking* [20]. To demonstrate this we developed a simple interface. A particular is shown in Fig.4b, relative to the extract of the counting grid shown in Fig.4a. The interface is *pannable* and *zoomable* and, at any moment, on the screen only the top $N = 500$ words are shown. To define the importance of each word in each position we weighted $\pi_{\mathbf{i}}(z)$ with the inverse document frequency. Fig.4b shows the lowest level of zoom: only words from few cells are visible and the font size resembles their weight. A user can zoom in to see the content of particular cells/areas, until he reaches the highest level of zoom when most of the words generated in a position are visible, Fig.4c.

We also propose a simple search strategy: once a keyword $\hat{z}$ is selected, each word $z$ in each position $\mathbf{j}$, is weighted with a word and position dependent weights. The first is equal to 1 if $z$ co-occur with $\hat{z}$ in some document, and 0 otherwise, while the latter is the sum of $\pi_{\mathbf{i}}(\hat{z})$ in all the $\mathbf{j}$s given that there exists a window $\mathbf{W_k}$ that contains both $\mathbf{i}$ and $\mathbf{j}$. Other strategies are of course possible. As result, this strategy highlights some areas and words, related to $\hat{z}$ on the grid and in each areas words related (similar topic) to $\hat{z}$ appears. Interestingly, if a search term is used in different contexts, few islands may appear on the grid. For example Fig.5 shows the result of the search for $\hat{z} =$"fry": The general frying is well separated from "deep frying" and "stir frying" which appears at the extremes of the same island. Presenting search results as islands on a

Figure 5: Search result for the word "fry".

2-dimensional grid, apparently improves the standard strategy, a linear list of hits, in which recipes relative to the three frying styles would have be mixed, while *tempura* have little to do with *pan fried noodles*.

## 4   Conclusion

In this paper we presented the componential counting grid model – which bridges the topic model and counting grid worlds – together with a similarity measure based on it. We demonstrated that the hidden mapping variables associated with each document can naturally be used in classification tasks, leading to the state of the art performance on a couple of datasets.

By means of proposing a simple interface, we have also shown the great potential of CCGs to visualize a corpora. Although the same holds for CGs, this is the first paper that investigate this aspect. Moreover CCGs subsume CGs as the components are used only when needed. For every restart, the grids qualitatively always appeared very similar, and some of the more salient similarity relationships were captured by all the runs. The word embedding produced by CCG has also advantages w.r.t. other Euclidean embedding methods such as ISOMAP [21], CODE [22] or LLE [23], which are often used for data visualization. In fact CCG's computational complexity is linear in the dataset size, as opposed to the quadratic complexity of [21, 21–23] which all are based on pairwise distances. Then [21, 23] only embed documents *or* words while CG/CCGs provide both embeddings. Finally as opposed to previous co-occurrence embedding methods that consider all pairs of words, our representation naturally captures the same word appearing in multiple locations where it has a different meaning based on context. The word "memory" in the Science magazine corpus is a striking example (memory in neruoscience, memory in electronic devices, immunologic memory).

## Footnotes

[1]Which may or may not be in the top three

[2]http://www.cs.cmu.edu/afs/cs.cmu.edu/project/theo-20/www/data/news20.html

[3]This happens for "reasonable" window sizes. For small windows (e.g, $2 \times 2$), the model doesn't have enough overlapping power and performs similarly a mixture model.

[4]We considered the following cuisines: *Afghan, Cajun, Chinese, English, French, German, Greek, Indian, Indonesian, Italian, Japanese, Mexican, Middle Eastern, Spanish and Thai*.

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
