[Supplementary Material 1]

onion add golden fry heat turmeric cumin coriander gravy chilli pressure cooker splutter wash aroma making airtight dry ingredient bag plastic place heavyduty large remain bowl small platter shake bacon small pour brains side scrape thick steak castiron thoroughly removable heat heat medium heat low high burn prevent vegetable heat medium onion saute

onion tomatoe garlic green add cauliflower floret pea fry chick cumin fry curry powder bhindi coconut curry jaggery veggie crush dry chappati malay gas drink wet processing processor workbowl metallic refrigeration ready container pesto bowl combination mousse motor pine nut toaste nut pine chipotle blender adobo puree blender cilantro chile bell occasional frequently stir occasional meatball low mixture stick beef beginning seasonal gallon croutons cabbage carrot celery celery crawfish green garlic onion garlic medhigh

cayenne salt garlic clove evaporation mace cinnamon clove partially pod peppercorn change cardamom low check mince two one cleaver one extreme knife steel stainless food processor hold glasses fit steel chill tea smoothe mound halfandhalf puree mill sit seeded pulverize puree chile margarine constantly stir low stir set panful spoonful reserve reserve porcini liquidizer pepper tomatoe parsley raise garlic

bay necessary evaporation add salt saffron rice pot biryani steam rice clarify basmati get chinese japanese lot possible chinese market purchase everything useful whisk mixture poach rectangular bitter salsa useful mexican lard tomatillo cupful verde thicken panful strain med glossy veal mussel quenelle oyster wine slowly has dilute paprika thyme

leaf ham meat necessary additional tenderizer absorbent fork cover meantime raisin water lid fluff grain rice cook double boiler gently frothy white brandy pour chocolate peak mascarpone gradual chocolate fold slowly sour shortening masa steamer refresh rinse asparagus boil water boil cold water boil hot five boil shell shellfish open any pound force earthenware sear flat bouillon

meat during chorizo continue level continue tenderizer adding slowly water enamel leave burner cupful dill clear dish run may reheat indian know going try useful container airtight electric meringue rum stiff beat yolk beat granulated clean bottom completion two stopping pot large quart firm drop water quart keep run clean slow dry crab lobster leg inside baste cognac before approximate dry

stove side wipe content point allowance leave base cutlet pour gently currant lowest tightfitting back full persian look needed completion tasty normal extract mixer vanilla speeds wire cake egg springform custard line hazelnut mix layer crumble lily flake according package immediately dente quart drain blanche drain baby fire clean puri fire sweetbread tail rub slit leg gravy burn needed pierce sharp

shank size fully teaspoonful teaspoonful spoonful carefully pour bottom wooden procedure spatula quickly round nonstick spread omelet flat flow least pancake texture biscotti griddle always crepe airtight long almond peach rack cinnamon sugar butter rind confectioners springing spread evenly evenly third tuna step pour pasta angel linguine sink cook colander drain tenderizer trim rinse separation stalk final hold eye clean body peach part live one tripe roast overcook kiwi forget careful papadum appalaam

allowance cool regularly panful larger side invert upside log apartment griddle cylinder spatula second cookie biscuit pretzel sift grease pie spread paper parchment golden fold pressing moisten ladyfinger egg crust assemble second third aside pesto sautpan basil instructions tossing til warm bite just grate transparent octopus oil mushy add way simple piece one eat piece bird during family excellence during thoroughly was call love since year error

polenta toaste mario anchovy soften stream place bread crumb panful olive fennel prosciutto crumb bread toothpick crumb crusty egg beaten lightly altogether omelet side slide slip turn diameter round inch together sheet egg pressing pastry egg finger set ricotta aside balsamic basil oregano basil spinach wilt tomatoe sundry garlic oil pepper black oil chunk capsicum squid rabbit pear greek summer knoedl was correct year list recipe basic recipe collection

olive wine rosemary rosemary rosemary seasonal fennel side salt side pour side brown dish mixture sheet additional beat sticky roll shape brush brush divide rectangle cut edge firmly divide seal edge overlap side filo corner one lay stack fashion marjoram steam alternative ounce feta olive oil herb caper parsley pepper stem outer size wash quarter okra easy home dish favorite early delicious restaurant people see include database basic see

yield calamari extravirgin smoke goldenbrown light olive divide oil degree piece saute fennel seasonal seasonal place removable chives place preheat minute ovenproof preheat middle sheet rack bowl turn mixer flour bulk dough knead board divide shape surface towel center form fill fold seal square triangle ravioli thaw opposite angle strip cut wide temperature room oct ahead zucchini scallion dill mint pita melon serve gently parsley flesh leave membrane peel horseradish dice small try core leave cut small white any preferable sprinkle suitable comment recipe cazuela came ask note recipe include note recipe sage course collection see basic

dredge platter thin excessive flour fryer batter tempura absorbent spoonful batter drop drizzle radicchio warm side flip beer pancake boondi plate bath drizzle three grapefruit fruit centre dessert enjoy set aside center zest pudding half completion cool segment four decorative custard dessert pecan dish cool insert tray tin pattern sharp resemblance cool soda squash honey cool walnut sugar bark grease cornmeal long moist pizza loaf surface form surface loose measurement soft pinch jam syrupy sweet scald puffed soft pinch ball palm bread feel has temperature five stone wet rawa double sprinkle rise starter size first little heap handful water effective another useful first another mash raw nonmetallic reddish spatter seafood crispy amount feeding neat fresh day several soak sure presoak dal overnight day month snail lima mung bamboo better before hour overnight hour before thoroughly dressings vinaigrette salad bean sprout best green beet nearly leftover ripe salad watercress sandwich keo cucumber slice crosswise reheat rock thinly individual diagonal arrange moment salad sauce oil quickly slice fry fried oil deepfry fry drain slice platter snow deep lift test golden brown wipe dip towel

one once sufficient glaze plus wheat pistachio banana sieve mould mango milk pineapple dissolve raisin jelly syrup flan apple dissolve caramelization hard proceeds cherry squeeze extract dhal cupful squeeze tamarind india coconut soon dried water crunchy become mung needed pulse smaller pinto stone accompaniment may similar jar week accompaniment pickle quince relish lettuce beds thin thin beef thin pork rib couple sesame soy tofu prawn oil fry cube second wok hot prawn swirl crisp hot

once one done matzoh passover three portion once quantity tray equal thick rose colouration film mold thread continue very eggplant springing torai very dry pulp variation size two five colorful darken hot thick piece split slightly become they give variety any may possible think flavorful flavorful preservative substitution sweet good bought optional msg peanut asian beef wok wok stirfry cornstarch mix cornstarch light coat apply tawa sizzler done

marination shallow afghan panfry grill refrigeration brush curd curdle muslin two curd least charcoal cloth paneer put pour leave semolina warm foil coarse chop baba panzerotti chop accommodate black tool pit turn oil another bitter chickpea bit turn chickpea mushy black kind find find one only quite eat state condiment cuisine root paste mexican ginger chow chinese stirfry sherry soya chestnut tbs scallion ginger appearance piece white cut

breast wings skin marinade sauerkraut save coal cheesecloth skewer tie charcoal outside wood thread possible useful leave length close fine min canning tblsp red lunch daal turn mediumlow oil optional wait taking once adjustable very dependent texture amount taste version guacamole mole style usual difference thai prik aromatic lime grass lemongrass indonesian vein noodle sauce aside set aside garlic bite

bone piece drip skim hour overnight marinade cavity carcasses turn secure preferable pattie smoker cube pack length cube away grate bottom chop tomatoe another grate half break sautthe moisture stew gold adjustable lose assemble taste plain proportion assemble tortilla enchilada enchilada extra tortilla choice say sound table tom curry chili coconut chili galangal shrimp devein devein pink chicken debone

duck cut turnip fat render excessive pork rib hare meat hour kabob turn stuffings push crock cube long order shredding crockpot jack refry uncooked spread spaghetti sauce pimiento cheese direction according vermicelli package macaroni sautfor cook mushroom sautee likely sauce completion individual wish slightly italian has serve dish meal soupe soupe tureen shallot mortar pestle sambal stock remain add removable add fryer dutch removable tongue

removable pot kidney large removable fat sausage goose cassoulet case stuff together spice mix stuff thoroughly ground lamb six burrito chipped taco uncover bubble spread mozzarella ricotta cheese parmesan sautonion sauce nutmeg sautuntil salt parsley wine parsley truffle tarragon good add dish linguine trout serve leftover provencal just limp immediately broth simmer soupe nectar intend reduce gentle bringing boil bringing boil return pot slotted

brown cook slightly salt cook pumpkin mixture yogurt spice dosa tava yogurt grinder ketchup rest ground paste microwaveable single electric powerful owner saint layer noodle lasagna cheese sauce pepper linguini parsley ovenproof wine parsley fernande garvin french dish tournedos warm serve cook french slowly broccoli cook kettle gumbo roux boil saucepan another saucepan bringing boil uncover simmer lower tenderizer

light onion set add potato golden add salt chile mustard jacket masala chile plain paste above paste mix grind mix paste apply remain right limit copyright repeat together hrs alternative dish melt spoonful margarine spray onethird butter dot flour consomme art flame source cap cook cast large moderator crisptender cook longer saucepan large foam minute reduce cover continue heat occasional cover gently saucepan soften simmer heat minute cover simmer heat lower tenderizer

light onion add onion golden add fry heat salt oil mustard seeded masala seeded fenugreek powder urad chaat dal chutney soak fine together ingredient foil twice together making fine dal pat broil broiler stand arrange combination remain combination hen large meanwhille small scallop dot sprinkle liver warm keep source hot beurre briskly minute skillet gizzards minute desirable continue heat minute cover saucepan reduce gently heat occasional cover add heat saute add minute

tomatoe oil fry heat mustard seeded masala powder seeded pop dal gram sambar fine wash dry making pat wrap aluminum broil every remain coat combination small turkey meanwhille large arrange surround small pour swordfish liver warm continue hot heavy heat cook medium high high uncover mediumsize large tenderizer mediumhigh nonstick tomatoe onion translucent onion

[Supplementary Material 2]

# Additional Material for the paper: Documents as multiple overlapping windows into grids of counts

## 1   Recipe Grid

In `AdditionalGrid.pdf` we reported the grid learned from the corpus of recipes that we have used as main example throughout all the paper (e.g., Fig. 1 and Fig. 4).

## 2   Additional Results

In `AdditionalResults.pdf` we reported the classification accuracy of CCG and LDA on 3 new Datasets. We used the same procedure described in Sec. 3-"Document Classification".

The first two plates (a-b), are relative to the `Different` and `Similar` subsets analyzed in the main text. Similarly to Fig. 3a of the main paper, on the x-axis we have the different model size, in term of capacity $\kappa$, whereas in the y-axis we reported the accuracy. The same $\kappa$ can be obtained with different choices of $\mathbf{E}$ and $\mathbf{W}$ therefore we represented the grid size $\mathbf{E}$ using gray levels, the lighter the marker the bigger the grid. The capacity $\kappa$ is roughly equivalent to the number of LDA topics as it represents the number of independent windows that can be fit in the grid and we compared the with LDA using this parallelism.

In the bottom plate, we have also considered the UIUC Sports dataset [1]. This computer vision dataset contains 8 classes of images each one relative to a different sport event. As words $z$ we extracted SIFT features and we quantized them in $Z = 200$ visual words. This dataset is particularly challenging as composing elements and objects must be identified in order to correctly classify the sport event. The rationale here is that different classes share some elements, like "water" for sailing and rowing classes, but they also will have peculiar elements that distinguish them.

## 3   Videos

We uploaded some video that presents a simple interface on youtube (anonymousNips channel). We could not compress the videos to fit the 10Mb limit due to the presence of text: high compression made the fonts unreadable.
In the videos we considered a new text corpus composed by all the Science Paper (1601 Reports or Research articles) published in 2011 and 2012. We extracted the words of each document from title, abstract, author names and references (we ignored the full text); this process resulted in a vocabulary size of 11038 unique words and a total of ∼2K tokens.

1. The interface is zoomable and pannable `http://www.youtube.com/watch?v=YUPj4iIy6zU`

2. Words are clickable `http://www.youtube.com/watch?v=oIvwC_6JwQ0`

3. Search for the word "Memory" `http://www.youtube.com/watch?v=ij2-XYUmhSI`

4. Refinement of a search by using additional search terms `http://www.youtube.com/watch?v=SmwDHrC-_tE`

5. Search for the word "Weather" `http://www.youtube.com/watch?v=60Yon4P-KYk`

6. Search for the word "Forest" `http://www.youtube.com/watch?v=oqM6uEDkqq8`

7. Words shown after a search changes based on the co-occurrence with the search term `http://www.youtube.com/watch?v=FF2a9WX1CrQ`

## References

[1] jia Li, L.: What, where and who? classifying event by scene and object recognition. In: In IEEE International Conference on Computer Vision. (2007)



[Supplementary Material 3]

# a) "Different"-20 NewsGroup Results

Classification Accuracy

Capacity κ / No. Topics

# b)"Similar"-20 NewsGroup Results

Classification Accuracy

Capacity κ / No. Topics

# c)  Sport Event Dataset (Visual features only) -
What, where and who? Classifying event by scene and object recognition, ICCV 2007

Classification Accuracy

Capacity κ / No. Topics

- Componential Counting Grid ($\theta$)
- Componential Counting Grid ($\Lambda$)
- LDA ($\theta$)
- Counting Grid ($q(l)$)