[Reviews · NeurIPS 2013]

Submitted by Assigned_Reviewer_4

The authors devise a new topic model that combines gradual topic changes of counting grids with a mixture model. As a byproduct the counting grid gives rise to new topic visualizations, as word distributions are embedded in a 2D plane. This paper evaluates the gains from giving the model the freedom to combine topics that are not close in the 2D space---in analog to a teleportation step. The new model is evaluated on three data sets.

The model extension is very reasonable, the derivation is correct. I deem the integration of counting grids into topic models as a valuable area of advancing topic models, that seemed to get stuck over the last few years.

The experimental evaluation on three data sets is rich with insight. The authors compares the model to a reasonable selection of related models, including counting grids without the extension.

The paper is well-written and understandable.

My only complaint is that the extension is rather straight-forward, where the main part of the paper is dedicated to introducing counting grids. On the other hand we appreciate the in-depth evaluation on the gains of this extension.

Summary: A reasonable extension to counting grid topic models. Convincing approach and experimental evaluation.

Submitted by Assigned_Reviewer_5

The paper presents an LDA-style mixed membership model where the component distributions are counting grids. Each document is a Dirichlet mixture over windows into the grid. Each window is a uniform distribution over the cells contained in that window. Each cell in the grid is a distribution over the vocabulary. The method is evaluated using classification accuracy, embedding quality, AUC in a retrieval task, and a visualization example.

I like this paper a lot. I think it's marginally above the accept threshold as is, and could be substantially improved with another round of revisions. I gave it an "incremental", but it's more like a 1.499.

The introduction does a reasonably good job presenting the basic intuition of the paper, that words often appear in a smooth continuum rather than distinct topics, and that the single-window model is too restrictive for real documents. Figure 1 is a bit of a missed opportunity. 1a works for me, but I found 1b and 1c confusing. A simpler toy example might be better, especially if it could be merged with Figure 2b.

I had a hard time moving from the introduction to section 2. In part, I felt there was a lot of technical terminology floating around, some of it redundant. What is a source, for example, or a bag? Are these cells and documents? I would pick a small set of concrete terms and use them consistently in both sections. Avoid "topic".

In Sect 2:

I would call pi_i(z) a distribution over the vocabulary, not a set of normalized counts. The latter may be its actual implementation in an algorithm, but here we're defining the model in the abstract. Give an example: pi_{1,1}("pizza") is the probability of the word "pizza" in the top left cell (or similar, if that's not correct)

What's a sample? Is that a document?

What is Z? The size of the vocabulary or the size of the grid? The variable z means "hidden variable" to me.

I'm really looking for a toy example.

Give an example for W, like 3x3.

Actually, I might restrict the initial presentation to square windows in 2D grids, for the sake of simplicity of notation, and then comment that the model is easily extended to d-dimensional grids and arbitrary window sizes.

I don't see why the number of non-overlapping windows is equivalent to the number of topics. The whole point is that the windows can overlap, no? Overlapping windows don't represent independent topics, but they do add modeling power, and that has to be taken into account for a fair comparison. Also, give a formula for kappa explicitly: it's not unambiguous right now (which ratio?).

I would rewrite section 2 around a step-by-step generative model. To generate a document, you first pick a distribution over windows. Then to generate each word, you pick a window, then pick a cell within the window, and finally pick a word from the cell.

The connection to LDA (windows of size 1x1) would be good to point out here.

It took a moment to figure out that U^W was a uniform distribution over a window. It looked like a matrix multiplication at first.

"Locations in the grid generate similar features" -- does this mean that the words in neighboring cells are semantically similar (but the words themselves differ), or that the distributions themselves are similar?

"The number of sources must not be specified..." First, I think this should be "need not be" -- "must not be" implies that you will get in trouble if you do! Second, I'm not sure what this means. The rest of the paper seems to assume that all dimensions are known in advance. Also, I'm still not sure what sources are (cells? windows?).

I didn't follow the marginalization step. I'd like a bit more explanation. If space is needed, variational update equations are an excellent candidate for supplementary material.

Distinguish between variables and distributions, and avoid referring to symbol alone: "the new updates for the cell distributions q(k) and the window distributions q(l) ..."

The comparison to Dunson et al is reasonable, but it might be worth talking about what is lost by using a grid rather than a continuous space.

Section 2.1 seems like a tangent.

Evaluation:

I didn't see held-out likelihood tests. It's not trivial for LDA. How would they be computed for this model?

I'd like to see a linear SVM used for document classification, in addition to the nearest neighbor method (I'm assuming that's BOW?).

See my earlier concerns about using "capacity" as a measure of approximate # of topics. Why not just the number of grid cells? In some sense CCG is just a fancier correlated distribution over the cells, like CTM or PAM.

The Euclidean embedding section is interesting. I'd like to see more of it.

What are the visual words? Are they given with the data set?

On the whole I buy the visualization aspect, because the word distributions are intrinsically embedded in a 2D space. I'd like to see more development of this idea. How would people use this embedding to learn something about the corpus? I didn't find Figure 5 particularly compelling for the fry/deep fry distinction, but the cluster of words that I associate with Indian recipes in the bottom right was interesting.
Summary: An interesting twist on topic models, with a more psychologically plausible notion of semantic gradients, and an intrinsic connection to visualization interfaces. Presentation is confusing, and results could be strengthened.

Submitted by Assigned_Reviewer_6

This paper describes a creative alternative for topic modeling:
mixed-membership on a "counting grid." The advantage of this approach
seems to be that you can move smoothly across the grid, achieving a
high effective number of topics while the spatial smoothing prevents
overfitting. The disadvantage seems to be that there are more
parameters (grid dimension and size, and window size). A variational
inference procedure that is somewhat to LDA is possible, although no
speed/complexity comparisons are provided. The spatial nature of the
approach has potential advantages for visualization as well.

One major problem for the paper is that the presentation is quite
confusing. This is partly due to the writing, which could generally
use a closer edit and more attention to detail. The mathematical
notation is also confusing, with too many variable names and some
inconsistency. For the probability of a word, we have four different
notations.

pi(z)
pi(w_n)
h_{i,z}
p(w_n | k_n, \pi)

I think the paper could survive with just two:
p_{W|K}(w | K = k) -- the probability of a word, conditioned on a draw directly from location k
p_{W|L}(w | L = l) -- the probability of a word, conditioned on a window centered on location l

with
p_{W|L}(w | L = l) \propto \sum_{k \in W_l} p_{W|K}(w | K = k) p_{K|L}(k | L = k),

where the last term has been defined to be uniform later in the paper
(this is not clear in Figure 2). Avoid capital I as a variable
name, especially in a sans-serif font.

The variational approximation seems to include a term q(\theta), but
this is never updated; rather, theta is updated directly in equation
(5). This may be connected with the definition of q(\theta) as a
"Dirac function centered at the optimal value theta_hat," but
theta_hat is never defined or updated. I think it would make more
sense just to treat this as a parameter and not bother to define
q(\theta).

The note that the "minimization procedure can be carried out
efficiently in O(n log n) using FFTs" is not helpful. At a minimum,
the reader needs a citation to a paper that explains how to do this;
but really, give specifics in an appendix. Also, please explain what
is n, and how it depends on the window size, vocabulary size, and
number of word tokens.

The marginalized updates (9) and (10) are presented in terms of the
overall likelihood rather than the variational bound. It is indeed
interesting that these updates don't depend on each other, although
it's a little misleading. If they were truly decoupled, there would be
no point in having l, as it would be decoupled from the observations
w. In fact, the update for q(l) affects theta, which in turn affects
the update for q(k); conversely, the update for q(k) affects \pi,
which in turn affects q(l).

The empirical results seem strong, beating published work on the 20
news task, and also on a new document classification datasets
(although the mind boggles that new document classification datasets
are still felt to be necessary). What is the meaning of the multiple
shadings of the circles in Figure 3? How were the model parameters
(grid size, dimensionality, window size) determined in the multimodal
retrieval and euclidean embedding experiments?

The visualization application seems interesting, although it seems
possible for individual words to appear in multiple positions, and
this has happened in Figure 5 for a few words (e.g.,
chicken). Wouldn't this be confusing for users?
Summary: Creative idea, confusing presentation.
Author Feedback

Author rebuttal: We thank the reviewers for positive scores and even much more positive comments. Since [3] is new and somewhat overlooked, we feel that this paper will serve both the purpose of popularizing the main idea of counting grids and making a significant step ahead in topic models, as the reviewers imply. This will make impact in various communities beyond ML such as NLP, Visualization, HCI, Neuroscience, CV.

*R1* Yes, Rome was not built in one day. Whenever a relatively new model is extended, part of the audience asks that the paper stands on its own, as the basic idea has not yet been digested. A significant fraction of time it is one of the extensions that finally alerts the community to the recent overlooked developments, and this makes it even more important for early papers to properly review the basic idea -- in this case [3]. So we explain this basic idea before moving to componential structure, which significantly enriches the model.

*R2* We will follow the suggestions to prepare the final version which will meet the expectations.
The overlapping windows (h at nearby locations) have similar feature distributions because they share many cells. pi can have variable neighboring distributions, though they do tend to be semantically similar.
Regarding the capacity, CCG has as many topics as cells in the grid since the window, regardless of its size could be positioned to start at any of the cells, each time yielding a new distribution over words. However, the differences for two highly overlapping windows are very small. So compared with LDA where topics are less constrained, CCG uses a much larger number of topics, but they only differ from their neighbors slightly. However, any LDA model can be turned into a CCG model of any size by simply designating large enough nonoverlapping windows to topics, filling them with words in one of many ways so that the sum (h) represents the topic distribution, and then using a prior over locations in the grid so that hybrid (overlapping windows) are forbidden. Thinking of a capacity of the grid as equal to the size of the grid would be a bit misleading, since we can train a 100X100 model on 100 datapoints without overfitting and even underfit when W=99X99. On the other hand, if the data is truly generated from an LDA model, the learned grid would, theoretically be learned to spread the topics into non-overlapping windows. Thus the notion of capacity as we use it in the paper k=|E|/|W|.
Lack of space prohibited further discussion of Eucl. embedding methods. For example, CCG’s embedding better preserves the original distances that are below a certain threshold: the correlation between the cosine distance between the counts and the symmetric KL divergence between theta’s (grid location posterior) is higher than the correlation with the Euclidean distance in the spaces provided by LLE and ISOMAP when the analysis is limited to document pairs with small distances (\rho=~0.4 vs ~0.3 after CE - all documents are embedded but we only look at how well small distances are preserved). This is especially important in visualization, where we mostly only care about small distances
Visual words are, as in [7], quantized sift descriptors extracted from the images.
In Tab.1, BOW is the result of linear SVM.
Regarding held-out likelihood, different models, even different learning algorithms, are hard to compare [9], and the comparisons often do not translate to performance differences in applications. In addition, CCGs subsume LDA, which means that some advantage for CCGs is expected by default in such tests.
Regarding the browsing applications, our experience shows that the grid consumption makes it much easier for us to formulate search expressions which filter the grid. This is because each word in the grid is accompanied by related words, stimulating users’ own thinking in terms of word associations (word embedding is close to mapping, so the user suddenly realizes that searching for mapping rather than embedding will find more journal papers related to our submission). Additionally docs that use the same word in different contexts are separated into different spatial clusters and this help disambiguation.
Finally, even if the user is after a relatively specific subset of documents, and though they can find them quickly using our interface, the user gets exposed to a variety of terms form other documents embedded nearby, serving a similar purpose that peripheral vision provides in vision: establishing context, and providing additional vague awareness of other related content that might be useful in the near future.

*R3* CCGs may seem to require more parameters be set than LDA, but this is not true in practice as once the W is “sufficiently big” (>3x3), the main parameter that matters is the ratio between grid and window areas. In Fig.3, the shaded circles indicate the CG area (see ln 300).
The fact that the same word can appear in multiple positions (if needed) is one of the points we didn’t emphasize enough! As opposed to previous co-occurrence embedding methods that consider all pairs of words, our representation naturally captures the same word appearing in different multi-word contexts. The word “memory” in the Science magazine corpus is a striking example (memory in neruoscience, memory in electronic devices, immunologic memory). For the most part, presenting a raw grid does make sense to the users once they get used to the presentation: a prominent word is prominent locally and derives one of its multiple meanings from its neighbors. The repetition of one word in nearby positions may seem odd, so we used a simple heuristic to prevent this.
In each task we employed the dataset author’s validation partition and protocol; if not available, we used 10% of the training data as validation set. We will clarify this in the introduction of Sec.5.
We thank R3 for the helpful suggestions regarding the notation and cleaning up the math section.